# Using big data to improve cardiovascular care and outcomes in China: a protocol for the CHinese Electronic health Records Research in Yinzhou (CHERRY) Study

Hongbo Lin,[1] Xun Tang,[2] Peng Shen,[1] Dudan Zhang,[2] Jinguo Wu,[3] Jingyi Zhang,[3] Ping Lu,[3] Yaqin Si,[2] Pei Gao[2]

► Prepublication history and additional material are available online. To view these files please visit the journal online (http://dx.doi.org/ 10.1136/bmjopen-2017-019698).

HL and XT contributed equally.

[1]Yinzhou District Center for Disease Control and Prevention, Ningbo, China
[2]Department of Epidemiology and Biostatistics, Peking University Health Science Center, Beijing, China
[3]Wonders Information Co.Ltd, Shanghai, China

**Correspondence to**
Professor Pei Gao;
peigao@bjmu.edu.cn

## ABSTRACT

**Introduction** Data based on electronic health records (EHRs) are rich with individual-level longitudinal measurement information and are becoming an increasingly common data source for clinical risk prediction worldwide. However, few EHR-based cohort studies are available in China. Harnessing EHRs for research requires a full understanding of data linkages, management, and data quality in large data sets, which presents unique analytical opportunities and challenges. The purpose of this study is to provide a framework to establish a uniquely integrated EHR database in China for scientific research.

**Methods and analysis** The CHinese Electronic health Records Research in Yinzhou (CHERRY) Study will extract individual participant data within the regional health information system of an eastern coastal area of China to establish a longitudinal population-based ambispective cohort study for cardiovascular care and outcomes research. A total of 1 053 565 Chinese adults aged over 18 years were registered in the health information system in 2009, and there were 23 394 deaths from 1 January 2009 to 31 December 2015. The study will include information from multiple epidemiological surveys; EHRs for chronic disease management; and health administrative, clinical, laboratory, drug and electronic medical record (EMR) databases. Follow-up of fatal and non-fatal clinical events is achieved through records linkage to the regional system of disease surveillance, chronic disease management and EMRs (based on diagnostic codes from the International Classification of Diseases, tenth revision). The CHERRY Study will provide a unique platform and serve as a valuable big data resource for cardiovascular risk prediction and population management, for primary and secondary prevention of cardiovascular events in China.

**Ethics and dissemination** The CHERRY Study was approved by the Peking University Institutional Review Board (IRB00001052-16011) in April 2016. Results of the study will be disseminated through published journal articles, conferences and seminar presentations, and on the study website (http://www.cherry-study.org).

### Strengths and limitations of this study

► The CHinese Electronic health Records Research in Yinzhou (CHERRY) Study is a large, natural population-based, observational cohort study linking big data of integrated individual-level electronic health records (EHRs).
► The CHERRY Study is among the first in China to establish a research platform from the EHR system for investigating a wide range of important issues regarding primary and secondary prevention of cardiovascular disease in a real world circumstance.
► The CHERRY Study is unique in its ability to trace the complete lifetime healthcare journey using birth certificates, health checks, primary care visits, hospitalisations, disease surveillance and ultimately, death certificates for one million adults in the general Chinese population.
► Missing data and conflicting data might be the main limitations of any EHR-converted big data research in terms of data quality; however, imputation for missing data within longitudinal cohorts and set-up of the priority of data sources for conflicting data may offer alternative solutions.
► Although the CHERRY Study has a relatively large number of participants, it is a regional cohort located in a developed area of China and as such, will not be nationally representative.

## INTRODUCTION

Cardiovascular disease (CVD) is the leading global disease burden worldwide.[1] Epidemiological studies have suggested that age-standardised CVD mortality rates have been falling in developed countries in recent decades.[2] However, China's current cardiovascular epidemic is increasing with rapid economic development and changing lifestyles.[3] Some established studies, such as the CArdiovascular disease research using LInked Bespoke studies and Electronic health Records

(CALIBER)[4] in the UK and CArdiovascular HEalth in Ambulatory care Research Team (CANHEART)[5] in Canada, have showed that forging a partnership between electronic health records (EHRs) and population-based cohort studies for CVD epidemiology is useful and growing. EHR-based data are rich with individual-level longitudinal measurement information. However, few existing studies in China to date have successfully assembled a cohort study based on EHRs at the population level. Success in linking big data to a population-based cohort could promote advances in improving cardiovascular outcomes and facilitating healthcare services research.[6]

Several critical factors affect the quality of EHR-based data for research, for example, appropriate approaches to data management and linkages, operational definitions for exposures, ascertainment and adjudication of outcomes, methods for handling missing data, and valid interpretation of clinical or public health integration and utility. These are all essential to generating robust findings.[7 8] Consequently, in this protocol, we aimed to describe the detailed methods and data sources used to establish an EHR-based general population-based cohort study in an eastern coastal area of China, the CHinese Electronic health Records Research in Yinzhou (CHERRY) Study. By linking various data sets at the individual level using unique and encoded identifiers, the CHERRY Study will be used as a research platform to address several important research questions, including but not limited to the following objectives:

1. What are the emerging risk factors, assembling both individual-level and community-level characteristics, for the incidence of major cardiovascular events in this developed area of China?
2. What is the up-to-date and suitable CVD risk assessment model to use for Chinese population in nutrition transition?
3. How do the different screening strategies for targeting people at high risk of CVD perform in a real world circumstance?
4. How does the regional health system deliver various CVD prevention practices in terms of cardiovascular outcome improvement?

## METHODS AND ANALYSIS

The CHERRY Study is a unique, population-based observational research resource that is aimed at improving cardiovascular care and outcomes in China. A particular focus will be on cardiovascular risk prediction and population management, providing evidence to improve the primary and secondary prevention of cardiovascular events.

## Study area

Yinzhou is a district of Ningbo in Zhejiang province (29°37′–29°57′ N, 121°08′–121°54′ E), located 230 km south of Shanghai, China, with an area of 1346 km$^2$ (figure 1). Yinzhou had a total population of 1.24 million

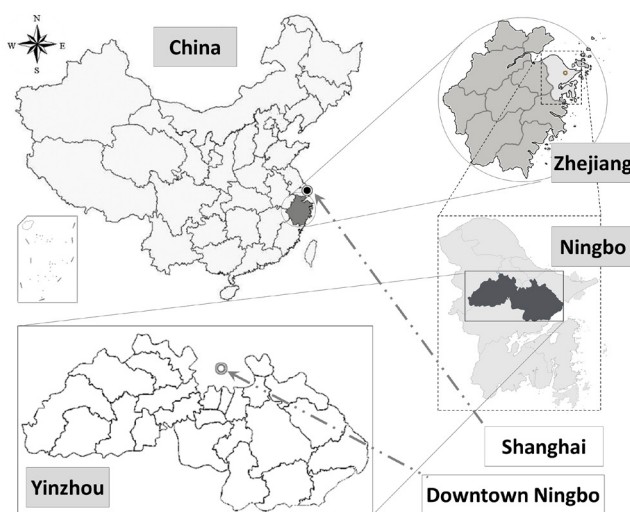

**Figure 1** Study location for the CHinese Electronic health Records Research in Yinzhou (CHERRY) study.

in 2016, including 790 600 permanent residents. This particular region is a developed area and was selected because of its outstanding general practice based primary care units and unique integrated electronic health information system. Particularly, the system has captured rich information from primary care units, including important information on cardiovascular risk factors and their disease management among communities. This enables us to use comprehensive big data for both epidemiological and clinical research in China.

## Data sources and cohort establishment

Health information systems in Yinzhou include different administrative databases of general demographic characteristics, health check information, inpatient and outpatient electronic medical records (EMRs), health insurance database, disease management and death certificates, and so on. The special feature of the system in Yinzhou is that these databases are inherently linked to each other by a unique and encoded identifier for each individual. The system was originally designed in 2006 by the Yinzhou District Centre for Disease Control and Prevention to facilitate routine primary care services for local general practitioners (GPs). It was then gradually integrated with information on public health surveillance, population screening, disease management, health information system in hospitals and other healthcare services. Since 2009, this regional system has covered nearly all health-related activities of residents within this region, from birth to death, including children, adolescents, pregnant women, adults and elderly people (figure 2 and online supplementary figure S1). Now 98% of permanent residents in Yinzhou have registered in the health information system with a valid healthcare identifier.

Consequently, based on the data sources in this integrated system, the CHERRY Study started in 2016 and is now extracting individual participants' data within the system to establish a natural, longitudinal, population-based,

**Research databases:**

**Data sources:**

**Figure 2** Data sources for establishing the CHinese Electronic health Records Research in Yinzhou (CHERRY) cohort. Notes: Although the main focus of the CHERRY Study is on adults, data sources of infants, children and pregnant women are included in the health information system, for example, birth weight from birth certificates. However, birth records are generally not available for adults who were already over 18 years old in 2009. Maternal exposures from antenatal examination records, that is, gestational hypertension or diabetes, were recorded in the system which can be potential risk factors for cardiovascular disease (CVD) prediction in women. However, a further ethics review process is required to extract maternal information in CHERRY.

ambispective cohort study for cardiovascular care and outcomes research. The study comprises all participants registered in the system if they: (1) were over 18 years of age on 1 January 2009; (2) have complete information on date of birth, sex and a valid healthcare identifier; (3) have been living in Yinzhou for at least 6 months; and (4) have Chinese nationality. We chose 1 January 2009 as the date of cohort inception to bypass the integration and preliminary test period of the system and to allow for full coverage of the regional population. Once included in the CHERRY cohort, individuals remain in it until death or termination of local health insurance (primarily owing to moving out of the province). Follow-up is generally continuous in the health information system. CHERRY will update certain important information such as vital status, clinical outcomes and claims data for all cohort members annually from the administrative databases. A third party, Wonders Information, was engaged to handle linkage and safe storage of the linked data sets, ensuring privacy protection in the CHERRY Study. A description of the CHERRY research cohort in relation to data sources in the administrative health information system is shown in figure 2 and online supplementary table S1. Each source captures a different aspect of a person's lifetime healthcare journey, as follows.

### Sociodemographics
Basic demographic and socioeconomic information of residents stems from the population census and registered health insurance database in the health information system. Key data variables will include date of birth, sex, ethnic group (eg, Han, Muslim), marital status, education, occupation and household information such as income, living space and so on.

### Longitudinal measurement of cardiovascular risk factors
Local GPs in Yinzhou have built up an impressive scheme on frequent health checks among adults and regular epidemiological surveys as part of primary care routine services over the 10 years after China's healthcare reform was initially launched.[9] According to the New Rural Cooperative Medical Scheme in China,[10] general health checks for adults in rural areas are conducted once every 2 years by local GPs. CHERRY then includes longitudinal measurements of risk factors related to CVD at the individual level, eg, smoking status, alcohol use, body mass index (BMI) and other obesity risk factors, and daily physical activity. Furthermore, screening of hypertension and diabetes mellitus in the general population aged 40 years and older is recommended by Chinese clinical practice guidelines and has been implemented in Yinzhou since 2009. Especially, lipid levels are frequently measured for the general population within the health check package paid for by employers, as part of employee welfare. Moreover, for adults with a history of hypertension or diabetes or individuals aged 60 years or over, a comprehensive medical examination is scheduled routinely at least once every year, for disease management. In total, 53% of individuals aged 40 years or over had at least one general health check information (eg, blood pressure measurements) within the system. Detailed information related to CVD risk factors within the examination is extracted to CHERRY, which includes blood measurements on glucose, haemoglobin A1c and lipid profiles (total cholesterol, high-density lipoprotein (HDL) cholesterol, low-density lipoprotein (LDL) cholesterol, triglycerides) as well as urine testing, and so on. Additional information from outpatient and inpatient EMRs will also be supplemented. Laboratory measurements in EMRs of circulating inflammation markers (eg, homocysteine, C reactive protein, albumin and leucocyte count), novel CVD related markers (eg, N-terminal pro B-type natriuretic peptide) or cardiovascular imaging information (eg, progression of coronary artery calcium) are included when available. The core variables for CVD-related factors and longitudinal measurements in the CHERRY study are listed in table 1.

### Healthcare services and medications
Outpatient and inpatient EMRs containing information of patients' healthcare services and medications will be transferred to CHERRY. EMR data contains physician visits, primary and secondary disease diagnosis, laboratory services, medications (indication, strength and dosage instructions) and so on from a network of five hospitals and over 289 primary care units across Yinzhou. The system (and CHERRY Study) contains EMRs in all the hospitals (both public and private hospitals) and primary care units within Yinzhou but no pharmacy stores. Therefore, the dispensation of medicine from both hospitals and primary care units is available (prescription medicines are always taken directly from the pharmacy within the hospitals/primary care units in China). Both individuals with and without health insurance can access the primary care and hospital services and therefore are all included in the system/study. By the end of 2015, 95.9% (1 010 658/1 053 565) adult participants with the unique identifier had EMRs in the system, receiving at least one clinical service (hospital or primary care). For patients receiving care outside Yinzhou (eg, patients might go to famous hospitals in Shanghai for certain complex surgical procedures), major non-fatal events occurred (eg, CVD and cancer) are tracked from both disease surveillance and chronic disease management systems. In this case, patients generally reported to local hospitals/GPs for aftersurgery health check services and drug prescription. Fatal events are tracked from the death registry where death certificates issued from hospitals outside Yinzhou are available. For the information related to primary disease diagnosis and medication used in hospitals outside the region, limited information is also extracted from the health insurance database. In Yinzhou, 95.7% of permanent residents are covered by the national health insurance. Residents of all ages were covered by the system. However, only adults above 18 years of age on 1 January 2009 are included in the CHERRY Study.

**Table 1** List of core risk factors for cardiovascular disease (CVD) in the CHinese Electronic health Records Research in Yinzhou (CHERRY) Study

| | Population census and registered health insurance database | Health checks database | Disease surveillance and management database | Outpatient/inpatient EMR database (including laboratory testing) | Charge and claims database | Environmental monitoring database |
|---|---|---|---|---|---|---|
| **Individual-level measurements** | | | | | | |
| Date of entry of first registration | * | | | | | |
| Date of the measurements | | † | † | † | † | |
| Date of birth, sex and ethnic groups | * | † | | † | † | |
| Marital status, education, occupation and socioeconomic status (household income, living space, etc) | * | | | † | | |
| Smoking and alcohol use (current/former/never; amount/duration, etc) | * | † | † | | | |
| Physical activity | * | † | † | | | |
| Weight, height, waist and hip circumference | * | † | † | | | |
| History of hypertension and diabetes mellitus | * | † | † | † | | |
| Prior history of coronary heart disease (in particular myocardial infarction and angina), stroke, transient ischaemic attack (TIA), and peripheral vascular disease (PVD) | * | † | † | † | | |
| Family history of diseases | * | † | † | | | |
| Systolic and diastolic blood pressure | * | † | † | | | |
| Metabolic factors (including fasting glucose, postload glucose and glycosylated haemoglobin) | | † | † | † | | |
| Lipid profiles: total, high-density and low-density lipoprotein cholesterol; triglycerides (including information about fasting status at time blood samples were taken) | | † | † | † | | |
| Blood urea nitrogen, creatinine, uric acid | | † | † | † | | |
| Use of cardiovascular medications (including antihypertensive drugs, 'statins', fibrates) and other medications (eg, hypoglycaemic agents, hormone replacement therapy) | | | † | † | † | |
| Inflammatory markers (including homocysteine, C reactive protein, fibrinogen, albumin, interleukin 6 and the leucocyte count) | | † | | † | | |
| Haemostatic factors (including von Willebrand factor and fibrin D-dimer) | | † | | † | | |

Continued

**Table 1** Continued

| | Population census and registered health insurance database | Health checks database | Disease surveillance and management database | Outpatient/inpatient EMR database (including laboratory testing) | Charge and claims database | Environmental monitoring database |
|---|---|---|---|---|---|---|
| Novel CVD-related markers (eg, NT-proBNP) | | | | † | | |
| ECG | | † | | † | | |
| Cardiovascular imaging information (eg, progression of coronary artery calcium, carotid intima media thickness) | | † | | † | | |
| Urine albumin to creatinine ratio (UACR) | | † | † | † | | |
| Cost of outpatient or inpatient admission (including fees for diagnosis, prescription, laboratory test, surgery, etc) | | | | † | † | |
| Environmental and ecological data | | | | | | |
| Date and region of surveillance | | | | | | † |
| Air temperature and precipitation | | | | | | † |
| Particles with aerodynamic diameter <2.5 µm (PM$_{2.5}$) | | | | | | † |
| Heavy metal concentration in water (lead, cadmium, mercury and arsenic) | | | | | | † |

\* indicates the database has only one record for the measurements.
† indicates the database has multiple records for the measurements.
Health checks database refers to health checks for new rural cooperative medical scheme, health checks for elderly people, and health checks for adults with hypertension and diabetes in figure 2.
EMR, electronic medical record; NT-proBNP, N-terminal pro B-type natriuretic peptide.

**Table 2** Definitions of major outcomes in the CHinese Electronic health Records Research in Yinzhou (CHERRY) Study

| Events of interest | ICD-10 code |
|---|---|
| **Primary events of interest** | |
| Death due to | |
| Ischaemic heart disease | I20–I25 |
| Cerebrovascular disease | I60–69 |
| Major cardiovascular disease | I00–I78 |
| All-cause mortality | |
| Hospitalisation with main diagnosis of | |
| Myocardial infarction | I21, I22 |
| Stroke | I60, I61, I63 (excluding I63.6), I64, H34.1 |
| Congestive heart failure | I50 |
| Outpatient visit with main diagnosis of | |
| Hypertension* | I10, I11, I12, I13, I15 |
| Diabetes mellitus† | E10, E11, E13, E14 |
| Cardiovascular disease | I00–I99 |
| **Secondary events of interest (fatal and non-fatal)** | |
| ST-elevation myocardial infarction (STEMI) | I21.0, I21.1, I21.2, I21.3, I22.0, I22.1, I22.8 |
| Non-ST segment elevation myocardial infarction (NSTEMI) | I21.4 |
| Ischaemic stroke | I63, I64, H34.1 (excluding I63.6) |
| Haemorrhagic stroke | I60, I61 |
| Transient ischaemic attack | G45 (excluding G45.4), H34.0 |
| Atrial fibrillation | I48 |
| Aortic aneurysm/aortic dissection | I71 |
| Peripheral artery disease | I70.2, I73.9, I74.3, I74.4 |

*Hypertension includes either registered in the hypertension management system, or self-reported history of hypertension during health checks, or hospital admission with a diagnosis of hypertension.
†Diabetes mellitus includes either registered in the diabetes management system, or self-reported history of diabetes during health checks or hospital admission with a diagnosis of diabetes.
Majority of ICD-10 codes were selected according to the published study: 'The Cardiovascular Health in Ambulatory Care Research Team (CANHEART): using big data to measure and improve cardiovascular health and healthcare services.' by Tu JV, *et al*, 2015, *Circulation: Cardiovascular Quality and Outcomes*, 8, p. 208. Copyright 2015 by the American Heart Association. Adapted with permission (License Number: 4252490297412). However, codes for STEMI and NSTEMI were modified based on the study in China.[34]
ICD-10, International Classification of Diseases, tenth revision.

## Clinical outcome events

For each individual, data will be sought on each of the following outcomes and their dates of occurrence: non-fatal myocardial infarction, non-fatal coronary heart disease (CHD), non-fatal stroke, cause-specific mortality (or at least fatal CHD and fatal stroke) and other cardiovascular outcomes. Precise details of the diagnostic criteria used to define cases have been sought from the 2014 American College of Cardiology/American Heart Association Key Data Elements and Definitions for Cardiovascular Endpoint Events in Clinical Trials.[11] Principal analyses will be based on events classified according to the International Classification of Diseases, tenth revision (table 2). In the CHERRY Study, for fatal outcomes, attribution of death refers to the primary cause provided by cause-specific mortality on death certificates in the health information system. Data undergo annual quality assessments. A description of the death certificates has been

reported previously.[12] For non-fatal outcomes, multiple sources exist in the system for the outcome definition, that is, disease management database (primary care), EMRs database (hospital care), health insurance database and disease surveillance database (disease registry). We define the disease surveillance database as the gold standard. In Yinzhou, CVD, hypertension, diabetes or cancer cases were required to be reported for disease surveillance and management by local GPs once the diagnoses were confirmed. Diagnosis of these diseases made by the physicians in all regional hospitals will be automatically sent to the local GPs of patients in the system. Criteria used for the diagnosis of incident cardiovascular morbidity in each source were described in online supplementary table S2. We define a 'definite' event if two or more sources excluding health insurance database report it as a case. A 'probable' event is defined if any source (including health insurance database) reports it

as a case. Cross-validation will be further investigated to improve the data quality and diagnostic validity. Primary and secondary prevention patients are defined as those without (primary) or with (secondary) a known history of CVD (table 2).

### Environmental and ecological characteristics

For environmental exposure monitoring data, we will include exposures to major water and air pollutants from eight environmental monitoring sites in Yinzhou, including heavy metal contamination and particles with aerodynamic diameter <2.5 μm ($PM_{2.5}$). These have been associated with increased cardiovascular mortality.[13 14] In addition, various meteorological conditions, such as air temperature and precipitation, from all weather stations across Yinzhou during the study period will also be available. Previous studies in China have demonstrated that both short-term (days) and longer-term (months or years) variations in temperature increase CVD morbidity and mortality.[15]

### Sample size

A total of 1 053 565 Chinese adults aged over 18 years were registered in the health information system. According to sample size requirements for prediction models,[16] for time to event outcomes, the number of participants experiencing the event must exceed 10 times the number of df, where the number of df includes the number of predictors screened for association with the outcome, all dummy variables, non-linear terms and interactions. By the end of 2015, there were 23 394 deaths (12 669 men and 10 725 women) and 14 217 cardiovascular events (7825 men and 6392 women) from 1 January 2009 to 31 December 2015. Thus, the sample size is generally sufficient for the CHERRY Study.

### Data analysis plan

A detailed data analysis plan will follow the checklist in the transparent reporting of a multivariable prediction model for individual prognosis or diagnosis (TRIPOD) guidelines[17] and recommendations by other studies.[16 18] In brief, data cleaning will proceed before examining exposure-outcome associations. Descriptive statistics will be used to determine values outside a plausible range, then outliers will be set to missing. Multiple imputation will be used to impute missing values for the predictors, where appropriate. CVD prediction models will be developed from sex-specific Cox proportional hazards models. The main CVD risk factors in established prediction models, such as Framingham Risk Scores,[19] will be retained in our model directly. We will then evaluate whether the predictive capability of the model will be improved by inclusion of additional predictors, using measures of discrimination and reclassification. The clinical performance of the models will be assessed by discrimination $C$ statistics, calibration $\chi^2$ and plots, Net Reclassification Improvement and Integrated Discrimination Improvement (IDI) indexes. Cross-validation will

be used to evaluate internal consistency; when available, the prediction models will also be evaluated for external validation in other independent cohort studies, such as the Fangshan Cohort Study.[20] All statistical analysis will be conducted using the SAS system, V.9.4 (SAS Institute, Cary, North Carolina, USA) and STATA software, V.14.1 (StataCorp, College Station, Texas, USA).

### DISCUSSION

The CHERRY Study is established to use longitudinal measurements of cardiovascular risk factors and disease prevention strategies in primary care among residents of Yinzhou, China. In practice, the Chinese CVD risk-assessment guidelines recommended a specific risk classification method based on the importance of risk factors on 10-year CVD risk, identified in two major cohort studies in China.[21] However, these cohorts were accrued decades ago and may not reflect the contemporary experience of the Chinese population. That is, the rapid economic transformation (industrialisation, marketisation, urbanisation and globalisation) in China has contributed to ageing populations, unhealthy lifestyles, environmental changes and epidemiological transitions.[3] In addition, although the recently published Prediction for atherosclerotic cardiovascular disease (ASCVD) Risk in China (China-PAR) model,[22] including traditional CVD risk factors for CVD prediction in Chinese, could be the potential tool, this has not been independently validated and not implemented in real clinical practice. Disparities in risk factor distributions, baseline survival and composition of disease subtypes were observed within China. Cut-offs to be used for 5-year and 10-year risk predictions in the Chinese population require more evidence from real world circumstances. The TRIPOD guidelines also recommended that prediction models using real world data should be developed.[17] We, therefore, aim to search for the up-to-date CVD risk assessment tool for CVD among the Chinese population under the current level of economic development in real world clinical practice settings. We will also provide evidence for different screening strategies in a real world circumstance. We expect that these data will be useful in supporting a wide range of cardiovascular epidemiology and public health research.

Linking big data from EHRs to a population-based cohort will be a powerful tool for investigating quality of care and improving cardiovascular outcomes. Big data studies in developed countries are generally robust. For example, CALIBER[4] in the UK is based on integrated healthcare databases with both nationwide EHRs in primary care and ongoing national quality registries; Sweden has similar health systems, but its primary care is organised regionally.[23] In addition, regional primary and ambulatory care data are also available for research linkages in the CANHEART study in Ontario, Canada.[5] Unfortunately, large-scale big data research on CVD that is based on EHRs is currently under-represented in

China.[24] Although CVD registries in China, such as the Chinese National Stroke Registry,[25] have been invaluable to drive research for secondary prevention of CVD, the evidence is often lacking for primary prevention in general populations. China's recent development in big data could facilitate EHR-based epidemiological studies of CVD, especially in some developed regions, such as Xiamen (Fujian Province) and Minhang District (Shanghai). However, little is known to understand the EHR landscape in China because of a lack of published protocols and low levels of data accessibility to these administrative healthcare databases.[26] The CHERRY Study has been inspired by all these studies, but differs in terms of its outstanding GP-based primary care units and unique integrated information system. To our knowledge,

the CHERRY Study is among the first in China to establish a research platform by linking big data across primary and secondary care and disease surveillance. In particular, it is unique in the ability to trace a complete lifetime healthcare journey in one million adults in a general Chinese population. CHERRY uses coded and inherently linked EHRs from primary care, hospitals, disease surveillance and ultimately, death registries. Particularly, many EHR-based cohorts in developed countries do not have complete information on lifestyle factors (eg, smoking status, alcohol use and others),[27] whereas 87% of the population has at least one measurement of smoking status or alcohol use in CHERRY (table 3).

It is known that less than 10 potentially modifiable risk factors might account for more than 90% of the

**Table 3** Characteristics of participants in the CHinese Electronic health Records Research in Yinzhou (CHERRY) Study, by sex

| | Men | | Women | | Overall | |
|---|---|---|---|---|---|---|
| | n=512043 | % having at least one measurement | n=541522 | % having at least one measurement | n=1053565 | % having at least one measurement |
| Age at first registration (years) | 40.17±15.17 | 100% | 39.24±15.56 | 100% | 39.69±15.38 | 100% |
| Region (n (%)) | | 98.93% | | 98.45% | | 98.68% |
| Rural | 354524 (69.99%) | | 364120 (68.30%) | | 718644 (69.12%) | |
| Urban | 152038 (30.01%) | | 169017 (31.70%) | | 321055 (30.88%) | |
| Education (n (%)) | | 79.05% | | 79.10% | | 79.07% |
| Junior high school or lower | 324438 (80.16%) | | 352534 (82.30%) | | 676972 (81.26%) | |
| Senior high school or higher | 80307 (19.84%) | | 75801 (17.70%) | | 156108 (18.74%) | |
| Smoking status (n (%)) | | 88.79% | | 85.60% | | 87.15% |
| Never smoker | 271870 (59.80%) | | 440834 (95.11%) | | 712704 (77.62%) | |
| Former smoker | 23099 (5.08%) | | 2828 (0.61%) | | 25927 (2.82%) | |
| Current smoker | 159664 (35.12%) | | 19861 (4.28%) | | 179525 (19.55%) | |
| Alcohol use (n (%)) | | 88.74% | | 85.57% | | 87.11% |
| ≥3 days per week | 87064 (19.16%) | | 3608 (0.78%) | | 90672 (9.88%) | |
| <3 days per week | 43811 (9.64%) | | 3713 (0.80%) | | 47524 (5.18%) | |
| Never drinker | 323518 (71.20%) | | 456059 (98.42%) | | 779577 (84.94%) | |
| Body mass index (BMI) (kg/m$^2$) | 22.71±2.32 | 86.97% | 22.25±2.57 | 84.06% | 22.47±2.46 | 85.47% |
| Systolic blood pressure (mm Hg) | 128.09±14.56 | 41.37% | 126.16±15.11 | 40.53% | 127.11±14.87 | 40.94% |
| Diastolic blood pressure (mm Hg) | 79.89±9.76 | 41.35% | 78.13±10.02 | 40.52% | 79.00±9.93 | 40.93% |

population-attributable risk of CVD worldwide.[28] [29] However, disparities in the effects of individual risk factors on CVD have also been found across different populations. Although the prevalence of CVD is declining in many developed countries with effective risk-lowering strategies for cardiovascular risk factors, such as smoking cessation or salt reduction, the prevalence of CVD in China is still increasing. The current focus on CVD prevention in the latest guidelines emphasises the use of risk assessment for appropriate prevention strategies aimed at those with a high risk of CVD.[30] This is consistent with the objectives of the CHERRY Study.

According to the checklist of the TRIPOD guidelines,[17] prediction models using real world data should be developed to discriminate the risk of death and events, and should be used in cost-effectiveness decision models for both public health and clinical practice. The CHERRY cohort will be established as a resource to encourage collaborative research based on individual participant data, so as to strengthen prognostic model development and external validation. For instance, recently published China-PAR models[22] for CVD prediction in Chinese populations could be validated in the CHERRY cohort. Furthermore, large-scale EHR-based cohort studies with detailed longitudinal measurements of clinical risk and patient care data over time are warranted, to better understand how risk factors influence the onset and progression of stroke and other CVDs. There are unique opportunities for the development of dynamic risk prediction models.[31] The CHERRY Study can be leveraged to develop risk prediction that is more organic, iterative and contemporary.

The study has some limitations. EHR data are known to suffer from a variety of data quality problems. Conflicting data across different sources in EHR-based data also exist in CHERRY. In the CALIBER Study,[32] the completeness and diagnostic validity of myocardial infarction recording varied across four EHR sources in primary care, hospital care, disease registry and mortality register. Thirty-one per cent of patients with non-fatal acute myocardial infarction were recorded in three out of four sources and 63.9% in at least two sources. Each data source missed a substantial proportion (25%–50%) of myocardial infarction events. A similar situation occurred in CHERRY. In addition, multiple records with similar but slightly different times of diagnosis for one patient may be recorded from different sources owing to varying timing accuracy. Prioritisation of sources in terms of conflicting data will be set up. Disease surveillance was considered as the gold standard. Events for one patient within a certain time range will be considered a single event; the allowed time window is disease-specific. Second, missing data are one of the main limitations of any EHR-converted research platform in terms of data quality. In CHERRY, data completeness varies (eg, 85.47% of people have at least one record on BMI measurement and 79.07% have their educational level recorded in the system (table 3)). Developments in imputation within longitudinal cohorts

may offer an alternative solution. Finally, although the study has a relatively large number of participants, it is a regional study that is located in a developed area of China. The study population is therefore not nationally representative.

## Ethics and dissemination

EHR use is becoming routine. Responsible data sharing is currently being defined, with principles established and policies set globally, such as the Health Insurance Portability and Accountability Act (HIPAA) and the Health Information Technology for Economic and Clinical Health (HITECH) Act in USA. Security, privacy, confidentiality and informed-consent issues are being carefully studied by many parties, and solutions are still in progress.[33] As China currently has not set its own standards nor defined implementation specifications and certification criteria for EHR use, as in the HIPAA and the HITECH Act, researchers in China can apply to the local health authority (Health and Family Planning Bureau of Yinzhou District) for information on EHR data for health research purposes, as well as seeking approval by institutional review boards based on international standards. For language and security reasons, foreign researchers are encouraged to apply through their Chinese partners, to facilitate international research collaborations. Although participants in the system are not provided with informed-consent as their information is routinely collected health data, the administrative data are inherently linked using unique encrypted identifiers to ensure privacy and confidentiality by the third-party company (Wonders Information). Results of the study will be disseminated through published journal articles, conferences and seminar presentations. More details will be published on the study website (http://www.cherry-study.org).

In summary, the CHERRY Study has the potential to provide population-based insights into the quality and outcomes of cardiovascular care. CHERRY will serve to decrease the burden of obtaining data in a research-ready format and encourage research collaboration.

**Acknowledgements** The authors thank the Health and Family Planning Bureau of Yinzhou District for providing access to the administrative databases used in the study. The authors also thank Analisa Avila, ELS, from Liwen Bianji, Edanz Group China (www.liwenbianji.cn/ac), for editing the English text of a draft of this manuscript.

**Contributors** HL, XT and PG drafted the manuscript. HL, XT, PS and PG conceived and designed the study. DZ, JW, JZ, PL and YS made substantial contributions to the study design. HL, JW and PS are responsible for study coordination; XT, PS and JZ are responsible for data quality control; DZ, PL and YS are responsible for data wrangling; XT, DZ, YS and PG are responsible for data analysis. All authors contributed to the writing of the study protocol in an iterative manner, and have read and approved the final manuscript.

**Funding** This study is supported by the National Natural Science Foundation of China (91546120, 81573226), Beijing Natural Science Foundation (7162107) and the National Thousand Talents Program for Distinguished Young Scholars, China (QNQR201501).

**Competing interests** None declared.

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
