## [Reviewer comments · BMJ Open]

ARTICLE DETAILS

TITLE (PROVISIONAL)	Using big data to improve cardiovascular care and outcomes in China: a protocol for the CHinese Electronic health Records Research in Yinzhou (CHERRY) study
AUTHORS	Lin, Hongbo; TANG, Xun; Shen, Peng; Zhang, Dudan; Wu, Jinguo; Zhang, Jingyi; Lu, Ping; Si, Yaqin; GAO, Pei

VERSION 1 – REVIEW

REVIEWER	Ellie Paige Australian National University, Australia
REVIEW RETURNED	29-Sep-2017

GENERAL COMMENTS	Overview: This paper presents the protocol for the integration and extraction of electronic health record data in the Yinzhou region of China. Such a dataset will provide a rich source of information on a broad range of CVD risk factors and includes integration with hospital and deaths records for ascertainment of fatal and non-fatal CVD events. There are several aspects of the protocol that would be improved by providing additional information and clarity. Main comments: 1. It would be useful to include in the introduction or discussion some context around how CVD risk is currently assessed in China. What CVD risk algorithms are used in practice and have these been validated for use in China?2. It's not clear whether this study has already been set-up and the data extracted or if it is in the planning stages. Please clarify this within the text.3. In Table 1, it would be helpful for the reader if the source of the measurements and the frequency of measurements were added to the table.4. The number of measurements people have depends on the number of times they visit their health provider and this is useful information to include when looking at CVD risk prediction using electronic health records. Is the number of contacts with a health provider going to be captured in the system?5. Information on other chronic diseases such as chronic kidney disease are sometimes included in CVD risk prediction equations. Will this information be captured in the system? Currently only diabetes is mentioned.6. Is there any way to identify related people within the system? Family history of CVD is a risk factor but it is not currently mentioned whether this can/will be captured.
--

	7. Little information is provided on the quality of the data recorded. It would be useful to include a more detailed discussion on this. 8. Some more information about the level of missing data would be useful – this is only briefly mentioned in the discussion. Would it be possible to include a table outlining the amount of missing data? 9. Conflicts in recording of risk factors between different data sources is mentioned as a limitation and it is stated that priority of the sources will be set up (pg. 14). How will this priority be decided in practice? Will it be based on the quality or completeness of the data source? 10. Cardiovascular medications are listed as being captured in the inpatient and outpatient electronic medical records. It would be useful to clarify what data will be captured. I.e. Does this include medicines prescribed in primary care and in hospital? Is there information on whether the medicines have been dispensed? Is there information on the indication, strength and dosage instructions for the medicines? Minor comments: 11. Some language checking and editing required. 12. Introduction, pg 5, states “Epidemiological studies have suggested that the morbidity and mortality of CVD in developed countries remain stable over past decades.” Age-standardised CVD mortality rates have been falling in developed countries (see: Roth, Gregory A., et al. "Global and regional patterns in cardiovascular mortality from 1990 to 2013." Circulation 132.17 (2015): 1667-1678.) 13. Will people moving into the region be captured in the study? This is not clear in the methods. 14. It is stated in the methods that only adults will be included study but birth records are also mentioned. Does this mean that birth records of the adults included in the system will be utilised or will pregnancy outcomes from the included adults be examined? Please make it clear how these records will be used. 15. For the sample size, the number of people experiencing the event is not clear so it is difficult to assess whether the sample size is sufficient 16. Discussion, pg. 12. It is not clear what is meant by the statement “We aim to search for the integrated risk assessment tools for cardiovascular disease in Chinese population under the current economic level of development.” 17. It would be useful for the reader if information was provided about whether the study data will be accessible to researchers, and if so, how it can be accessed.
--	---

REVIEWER	Jack Tu Institute for Clinical Evaluative Sciences Sunnybrook Schulich Heart Centre University of Toronto Canada
REVIEW RETURNED	11-Oct-2017

GENERAL COMMENTS	This is a clearly written manuscript describing a new CHERRY electronic medical health records cohort in Yinzhou china. The level of English is pretty good but could be strengthened further by having an English-editor rewrite some of the sentences.
--

	1) The databases appear to be fairly comprehensive but it not clear what percentage of clinical care that the patients receive is covered in the database (?100%, 95%, etc.) Are there any opportunities for patients to receive care outside of the EHR network. Are all ages covered for all the databases mentioned (drugs, hospitalizations, health checks). What about patients without health insurance coverage? What percentage of the population has health insurance? 2) Have any studies been done to cross-validate any of the information against external gold-standard sources (e.g. chart reviews). 3) Could the authors comment on privacy and security issues along with research ethnics board oversight. Are the authors allowed to used this information for health research purposes? Are patients informed about what happens to their information? How is the data linked and deidentified to ensure patient confidentiality. 4) It would be interesting if the authors could include a table showing some baseline data from their cohort, along with the rates of missing data? 5) Many EHR cohorts in the west don't have complete information on lifestyle factors (e.g. smoking, diet. etc.). How good will CHERRY be in this regards? 6) More discussion of how this cohort compares with other similar cohorts in China would be of interest. Is this the only region with such integrated data systems or are there other regions of China developing similar systems.
--	--

VERSION 1 – AUTHOR RESPONSE

Reviewer 1

Overview: This paper presents the protocol for the integration and extraction of electronic health record data in the Yinzhou region of China. Such a dataset will provide a rich source of information on a broad range of CVD risk factors and includes integration with hospital and deaths records for ascertainment of fatal and non-fatal CVD events. There are several aspects of the protocol that would be improved by providing additional information and clarity.

Main comments:

1. It would be useful to include in the introduction or discussion some context around how CVD risk is currently assessed in China. What CVD risk algorithms are used in practice and have these been validated for use in China?

Response: We have now stated in the discussion section (page 13, line 14) as “In practice, the Chinese CVD risk-assessment guidelines recommended a specific risk classification method based on the importance of risk factors on 10-year CVD risk, identified in two major cohort studies in China [Nature Reviews Cardiology 2015;12(5):301-11.].

However, these cohorts were accrued decades ago and may not reflect the contemporary experience of Chinese population. That is, the rapid economic transformation (industrialization, marketization, urbanization, and globalization) in China has contribute to aging populations, unhealthy lifestyles, environmental changes, and epidemiological transitions [Circulation 2016;133(24):2545-60]. In addition, although recently published China-PAR (Prediction for ASCVD Risk in China) model [Circulation 2016;134(19):1430-40], including traditional CVD risk factors for CVD prediction in Chinese, could be the potential tool, this has not been independently validated and not implemented in real clinical practice. Disparities on risk factor distributions, baseline survival and composition of disease subtypes were observed within China. Cutoffs to be used for 5- and 10-year risk predictions in Chinese population require more evidence from real-world circumstances. The TRIPOD guidelines also recommended that prediction models using real-world data should be developed [BMJ 2015;350:g7594]. We therefore aim to search for the up-to-date CVD risk assessment tool for cardiovascular disease among a Chinese population under the current level of economic development in real-world clinical practice settings.”

2. It's not clear whether this study has already been set-up and the data extracted or if it is in the planning stages. Please clarify this within the text.

Response: We clarified that CHERRY study started in 2016 and the data extraction is ongoing in the Methods section (page 7, line 22): “CHERRY study started in 2016 and is now extracting individual participants' data within the system to establish a natural, longitudinal, population-based, ambispective cohort study for cardiovascular care and outcomes research.”

3. In Table 1, it would be helpful for the reader if the source of the measurements and the frequency of measurements were added to the table.

Response: As suggested, the data sources and frequency of measurements are now added to the new Table 1 (page 23).

4. The number of measurements people have depends on the number of times they visit their health provider and this is useful information to include when looking at CVD risk prediction using electronic health records. Is the number of contacts with a health provider going to be captured in the system?

Response: Yes. As we illustrated in the Table 1 (page 23), all individual-level records included the information on date of visit, which are recorded in the system (and extracted to CHERRY study). We could count the number of contacts per person.

5. Information on other chronic diseases such as chronic kidney disease are sometimes included in CVD risk prediction equations. Will this information be captured in the system? Currently only diabetes is mentioned.

Response: Yes. As we have listed in the Table 1 (page 23), blood urea nitrogen, albumin, uric acid, creatinine, and Cystatin C are captured from both health checks and inpatients/outpatients EMRs in the system and extracted to CHERRY study. Estimated glomerular filtration rate (GFR) and albumin-to-creatinine ratio (ACR) could be calculated as information related to chronic kidney disease.

6. Is there any way to identify related people within the system? Family history of CVD is a risk factor but it is not currently mentioned whether this can/will be captured.

Response: In the registered health insurance database, the relationship could be identified by the FamilyID and RELATION (Dataset code: KTHRA_HEALTH_ARCHIVES). Moreover, family history of CVD is also collected through the registered health insurance database (Dataset code: KTHRA_FAMILY_HISTORY) and the specific variable in disease management and surveillance (Dataset codes: KTCEDMS_CARD & KTCEDMS_HIGHRISK_CARD). Please see the Supplemental Files Table S1 for the description of dataset sources and code used in CHERRY.

7. Little information is provided on the quality of the data recorded. It would be useful to include a more detailed discussion on this.

Response: We thank the Reviewer for this suggestion. We have now provided more information related to the quality of the data, conflicting data and missing data in the manuscript. In the Method section (page 11, line 7 and line 14), we now stated “multiple sources exist in the system for the outcome definition, i.e., disease management database (primary care), EMRs database (hospital care), health insurance database, and disease surveillance database (disease registry). We define the disease surveillance database as gold standard. Criteria used for the diagnosis of incident cardiovascular morbidity in each source were described in Table S2. We define a “definite” event if two or more sources excluding health insurance database reported as a case. A “probable” event is defined if any source (including health insurance database) reported as a case. Cross-validation will be further investigated to improve the data quality and diagnostic validity”. In addition, in the discussion (page 16, line 7 and line 20) we clarified that “EHR data are known to suffer from a variety of data quality problems. Conflicting data across difference sources in EHR-based data also exist in CHERRY. In CALIBER study [BMJ 2013;346:f2350], the completeness and diagnostic validity of myocardial infarction recording varied across four EHRs sources in primary care, hospital care, disease registry, and mortality register. 31.0% of patients with non-fatal acute myocardial infarction were recorded in three out of four sources and 63.9% in at least two sources. Each data source missed a substantial proportion (25-50%) of myocardial infarction events. Similar situation occurred in CHERRY. ...Secondly, missing data is one of the main limitations of any EHR-converted research platform in terms of data quality. In CHERRY, data completeness varies [e.g., 85.47% of people have at least one record on body mass index (BMI) measurement and 79.07% have their educational level recorded in the system (Table 3)]. Developments in imputation within longitudinal cohorts may offer an alternative solution.”

8. Some more information about the level of missing data would be useful – this is only briefly mentioned in the discussion. Would it be possible to include a table outlining the amount of missing data?

Response: As suggested, the new Table 3 (page 26) outlining the information of certain risk factors as well as the completeness of data is now included. Please also refer to the response above.

9. Conflicts in recording of risk factors between different data sources is mentioned as a limitation and it is stated that priority of the sources will be set up (pg. 14). How will this priority be decided in practice? Will it be based on the quality or completeness of the data source?

Response: As described above, we now clarified in the Method section (page 11, line 7 and line 14) and provided the new Table S2 (Supplemental files) as “multiple sources exist in the system for the outcome definition, i.e., disease management database (primary care), EMRs database (hospital care), health insurance database, and disease surveillance database (disease registry). We define the disease surveillance database as gold standard. Criteria used for the diagnosis of incident cardiovascular morbidity in each source were described in Table S2. We define a “definite” event if two or more sources excluding health insurance database reported as a case. A “probable” event is defined if any source (including health insurance database) reported as a case. Cross-validation will be further investigated to improve the data quality and diagnostic validity”.

10. Cardiovascular medications are listed as being captured in the inpatient and outpatient electronic medical records. It would be useful to clarify what data will be captured. I.e. Does this include medicines prescribed in primary care and in hospital? Is there information on whether the medicines have been dispensed? Is there information on the indication, strength and dosage instructions for the medicines?

Response: We now clarified in the Methods section (page 9, line 24) as “EMR data contains physician visits, primary and secondary disease diagnosis, laboratory services, medications (indication, strength and dosage instructions) and so on from a network of 5 hospitals and over 289 primary care units across Yinzhou. The system (and CHERRY study) contained EMRs in all the hospitals (both public and private hospitals) and primary care units within Yinzhou but no pharmacy stores. Therefore, dispensation of medicine from both hospitals and primary care units are available (prescription medicines are always taken directly from the pharmacy within the hospitals/primary care units in China). Both individuals with and without health insurance can access the primary care and hospital services and therefore are all included in the system/study.”

Minor comments:

11. Some language checking and editing required.

Response: As also suggested by the Editor, the revised manuscript has been proofread by Edanz Editing Services for error check in language.

12. Introduction, pg 5, states “Epidemiological studies have suggested that the morbidity and mortality of CVD in developed countries remain stable over past decades.” Age-standardised CVD mortality rates have been falling in developed countries (see: Roth, Gregory A., et al. "Global and regional patterns in cardiovascular mortality from 1990 to 2013." *Circulation* 132.17 (2015): 1667-1678.)

Response: Revised as suggested (page 5, line 3).

13. Will people moving into the region be captured in the study? This is not clear in the methods.

Response: Though further update may be applied, we do not include people moving into the region after 1 January 2009 at the current stage according to the inclusion criteria in CHERRY (Page 7, line 25). We have clarified this in the Methods section.

14. It is stated in the methods that only adults will be included study but birth records are also mentioned. Does this mean that birth records of the adults included in the system will be utilised or will pregnancy outcomes from the included adults be examined? Please make it clear how these records will be used.

Response: We thank the reviewer for this comment. Because the system was firstly designed in 2006, birth records are generally not available for adults who were over 18 years old in 2009. However, pregnancy outcomes for women, i.e. gestational hypertension or diabetes, were recorded in the system which can be potential risk factors for CVD prediction in women. However, further ethics review process is required to extract maternal information in CHERRY. That's why this information was grayed in Figure 2 at the current stage but could be probably added in the future. We have clarified this information in the footnote of Figure 2 (page 25).

15. For the sample size, the number of people experiencing the event is not clear so it is difficult to assess whether the sample size is sufficient.

Response: We clarified in the Methods-sample size section (page 12, line 12) that “By the end of 2015, there were 23,394 deaths (12,669 men and 10,725 women) and 14,217 cardiovascular events (7,825 men and 6,392 women) from 1 January 2009 to 31 December 2015. Thus, the sample size is generally sufficient for the CHERRY study.”

16. Discussion, pg. 12. It is not clear what is meant by the statement “We aim to search for the integrated risk assessment tools for cardiovascular disease in Chinese population under the current economic level of development.”

Response: We have now clarified this in the Discussion section (page 13, line 19). Please refer to the response #1.

17. It would be useful for the reader if information was provided about whether the study data will be accessible to researchers, and if so, how it can be accessed.

Response: We now stated in the Ethics and dissemination section (page 17, line 7) as “As China currently has not set its own standards nor defined implementation specifications and certification criteria for EHR use, as in the HIPAA and the HITECH Act, researchers in China can apply to the local health authority (Health and Family Planning Bureau of Yinzhou District) for information on EHR data for health research purposes, as well as seeking approval by institutional review boards (IRBs) based on international standards. For language and security reasons, foreign researchers are encouraged to apply through their Chinese partners, to facilitate international research collaborations. ... More details will be published on the study website (<http://www.cherry-study.org>).”

Reviewer 2

This is a clearly written manuscript describing a new CHERRY electronic medical health records cohort in Yinzhou china. The level of English is pretty good but could be strengthened further by having an English-editor rewrite some of the sentences.

1) The databases appear to be fairly comprehensive but it not clear what percentage of clinical care that the patients receive is covered in the database (?100%, 95%, etc.)

Are there any opportunities for patients to receive care outside of the EHR network.

Are all ages covered for all the databases mentioned (drugs, hospitalizations, health checks). What about patients without health insurance coverage? What percentage of the population has health insurance?

Response: We thank the reviewer for this comment. We now clarified in the Methods section (page 9, line 28) as “The system (and CHERRY study) contained EMRs in all the hospitals (both public and private hospitals) and primary care units within Yinzhou but no pharmacy stores. Therefore, dispensation of medicine from both hospitals and primary care units are available (prescription medicines are always taken directly from the pharmacy within the hospitals/primary care units in China). Both individuals with and without health insurance can access the primary care and hospital services and therefore are all included in the system/study. By the end of 2015, 95.9% (1,010,658 / 1,053,565) adult participants with the unique identifier had EMRs in the system, receiving at least one clinical service (hospital or primary care). For patients receiving care outside Yinzhou (e.g., patients might go to famous hospitals in Shanghai for certain complex surgical procedures), major non-fatal events occurred (e.g., CVD and cancer, etc) are tracked from both disease surveillance and chronic disease management system. In this case, patients generally reported to local hospital/GPs for after-surgery health check services and drug prescription. Fatal events are tracked from death registry where death certificates issued from hospitals outside Yinzhou are available. For the information related to the primary disease diagnosis and medication used in hospitals outside the region, limited information is also extracted from the health insurance claims database. In Yinzhou, 95.7% of permanent residents are covered by the national health insurance. Residents of all ages were covered within the system. However, only adults above 18 years of age on 1 January 2009 are included in the CHERRY study.”

2) Have any studies been done to cross-validate any of the information against external gold-standard sources (e.g. chart reviews).

Response: we now clarified in the Method section (page 11, line7 and line 14) and provided the new Table S2 (Supplemental files) as “multiple sources exist in the system for the outcome definition, i.e., disease management database (primary care), EMRs database (hospital care), health insurance database, and disease surveillance database (disease registry).

We define the disease surveillance database as gold standard. Criteria used for the diagnosis of incident cardiovascular morbidity in each source were described in Table S2. We define a “definite” event if two or more sources excluding health insurance database reported as a case. A “probable” event is defined if any source (including health insurance database) reported as a case. Cross-validation will be further investigated to improve the data quality and diagnostic validity”. In addition, in the discussion (page 16, line 7) we have also stated in the limitations that “EHR data are known to suffer from a variety of data quality problems. Conflicting data across different sources in EHR-based data also exist in CHERRY. In CALIBER study [BMJ 2013;346:f2350], the completeness and diagnostic validity of myocardial infarction recording varied across four EHRs sources in primary care, hospital care, disease registry, and mortality register. 31.0% of patients with non-fatal acute myocardial infarction were recorded in three out of four sources and 63.9% in at least two sources. Each data source missed a substantial proportion (25-50%) of myocardial infarction events. Similar situation occurred in CHERRY.”

3) Could the authors comment on privacy and security issues along with research ethics board oversight. Are the authors allowed to use this information for health research purposes? Are patients informed about what happens to their information?

How is the data linked and deidentified to ensure patient confidentiality.

Response: We thank the reviewer for this comment. We incorporated the reviewer’s points in the Ethics and dissemination section (page 17, line 2) as “EHR use is becoming routine. Responsible data sharing is currently being defined, with principles established and policies set globally, such as the Health Insurance Portability and Accountability Act (HIPAA) and the Health Information Technology for Economic and Clinical Health (HITECH) Act in the United States. Security, privacy, confidentiality, and informed-consent issues are being carefully studied by many parties, and solutions are still in progress [N Engl J Med 2014;370(23):2163-5]. As China currently has not set its own standards nor defined implementation specifications and certification criteria for EHR use, as in the HIPAA and the HITECH Act, researchers in China can apply to the local health authority (Health and Family Planning Bureau of Yinzhou District) for information on EHR data for health research purposes, as well as seeking approval by institutional review boards (IRBs) based on international standards. For language and security reasons, foreign researchers are encouraged to apply through their Chinese partners, to facilitate international research collaborations. Although participants in the system are not provided with informed-consent as their information is routinely collected health data, the administrative data are inherently linked using unique encrypted identifiers to ensure privacy and confidentiality by the third-party company (Wonders Information Co., Ltd.)”

4) It would be interesting if the authors could include a table showing some baseline data from their cohort, along with the rates of missing data?

Response: As suggested, the new Table 3 (page 26) outlining the information about certain risk factors in baseline along with the completeness of data is now included in the manuscript.

5) Many EHR cohorts in the west don't have complete information on lifestyle factors (e.g. smoking, diet. etc.). How good will CHERRY be in this regards?

Response: We thank the Reviewer for this suggestion. The new Table 3 outlining the information of certain lifestyle risk factors as well as the completeness of data is now included (page 27). In the discussion (page 16, line 22) we clarified that “In CHERRY, data completeness varies [e.g., 85.47% of people have at least one record on body mass index (BMI) measurement and 79.07% have their educational level recorded in the system (Table 3)]. Developments in imputation within longitudinal cohorts may offer an alternative solution.”

6) More discussion of how this cohort compares with other similar cohorts in China would be of interest. Is this the only region with such integrated data systems or are there other regions of China developing similar systems.

Response: We thank the Reviewer for this suggestion. In discussion (page 14, line 21), we now described that “China's recent development in big data could facilitate EHR-based epidemiological studies of CVD, especially in some developed regions, such as Xiamen (Fujian Province) and Minhang District (Shanghai). However, little is known to understand the EHR landscape in China because of a lack of published protocols and low levels of data accessibility to these administrative healthcare databases [J Mark Access Health Policy 2015;3.]. The CHERRY study has been inspired by all these studies, but differs in terms of its outstanding GP-based primary care units and unique integrated information system.”

VERSION 2 – REVIEW

REVIEWER	Ellie Paige Australian National University, Australia
REVIEW RETURNED	07-Dec-2017

GENERAL COMMENTS	The authors have sufficiently addressed my previous comments and the manuscript has been substantially improved with their changes.
---

REVIEWER	Jack Tu Institute for Clinical Evaluative Sciences Sunnybrook Schulich Heart Centre University of Toronto Canada
REVIEW RETURNED	17-Dec-2017

GENERAL COMMENTS	The authors have done a very good job of responding to the reviewers comments and the level of English has been improved. Table 2 appears to have adapted from a previous CANHEART publication by myself in Circulation Quality and Outcomes 2015:8: 204-12. If so, this should be explicitly acknowledged. Some of these ICD-codes are from a Canadian Adaption of ICD-10 (eg. R codes for STEMI) and I am uncertain if they are used in China. This should be clarified.
---

VERSION 2 – AUTHOR RESPONSE

Reviewer: 1

Reviewer Name: Ellie Paige

Institution and Country: Australian National University, Australia

Please state any competing interests or state 'None declared': None declared

Please leave your comments for the authors below

The authors have sufficiently addressed my previous comments and the manuscript has been substantially improved with their changes.

Response: We appreciate the Reviewer's comments of support.

Reviewer: 2

Reviewer Name: Jack Tu

Institution and Country: Institute for Clinical Evaluative Sciences, Sunnybrook Schulich Heart Centre, University of Toronto, Canada

Please state any competing interests or state 'None declared': None Declared

Please leave your comments for the authors below

The authors have done a very good job of responding to the reviewers' comments and the level of English has been improved.

Table 2 appears to have adapted from a previous CANHEART publication by myself in *Circulation Quality and Outcomes* 2015;8: 204-12. If so, this should be explicitly acknowledged. Some of these ICD-codes are from a Canadian Adaption of ICD-10 (eg. R codes for STEMI) and I am uncertain if they are used in China. This should be clarified.

Response: We thank the Reviewer for this comment. In Table 2 (page 25), we now described that "Majority of ICD-10 codes were selected according to the published study: "The Cardiovascular Health in Ambulatory Care Research Team (CANHEART): using big data to measure and improve cardiovascular health and healthcare services." by Tu JV, et al., 2015, *Circulation: Cardiovascular Quality and Outcomes*, 8, p. 208. Copyright 2015 by the American Heart Association, Inc. Adapted with permission (License Number: 4252490297412). However, codes for the STEMI and NSTEMI were modified based on the study in China. [*Medicine (Baltimore)*. 2016; 95(5):e2677]". We have clarified this information in the footnote of Table 2 (page 25).